# Tear Fluid Biomarkers in Diabetic Ocular Surface Disease: A Systematic Review

**DOI:** 10.3390/jcm14196958

**Published:** 2025-10-01

**Authors:** Natalia Gospodarczyk, Anna Martyka, Urszula Błaszczyk, Wiktoria Czuj, Julia Piekarska, Edward Wylęgała, Anna Nowińska

**Affiliations:** 1Department of Ophthalmology, District Railway Hospital in Katowice, Medical University of Silesia, 40-760 Katowice, Poland; aniamartyka98@gmail.com (A.M.); czujwiktoria@gmail.com (W.C.); julia.aleksandra.piekarska@gmail.com (J.P.); ewylegala@sum.edu.pl (E.W.); anna.nowinska@sum.edu.pl (A.N.); 2Department of Ophthalmology, Faculty of Medical Sciences in Zabrze, Medical University of Silesia, 40-760 Katowice, Poland; 3Department of Biochemistry, Faculty of Medical Sciences in Zabrze, Medical University of Silesia, 40-760 Katowice, Poland; ublaszczyk@sum.edu.pl

**Keywords:** biomarkers, tear fluid, diabetes, diabetic ocular surface disease

## Abstract

**Background:** Diabetic eye surface disease, including dry eye syndrome, corneal neuropathy, and diabetic retinopathy, is a common complication of diabetes. Tear fluid biomarkers may aid in early diagnosis and disease monitoring. The objective of this systematic review was to identify and evaluate tear fluid biomarkers in diabetic ocular surface disease according to PRISMA guidelines. **Methods:** PubMed, Scopus, and Embase databases were searched through June 2025. Eligible studies included clinical and observational studies measuring proteins, lipids, cytokines, trace elements, or nucleic acids in tear fluids in patients with diabetes. **Results:** The search identified 198 studies, and of those, 30 studies were included, comprising 14 original investigations with 871 participants (133 with type 1 diabetes, 453 with type 2 diabetes, 16 with pre-diabetes, and 269 healthy controls). The main biomarker categories were cytokines (IL-6, IL-8, TNF-α, and MMP-9), neuropeptides (substance P, NPY), proteins (IGFBP-3, progranulin), lipids, glycans, microRNAs, circRNAs, and trace elements. **Conclusions:** More than a dozen biomarkers in the tear fluid have been identified that may reflect diabetes-related changes in the ocular surface. Tear fluid analysis may be a valuable tool in personalizing the diagnosis and treatment of diabetic ocular surface diseases, but further studies are needed to confirm its clinical significance.

## 1. Introduction

Diabetes mellitus (DM) is a chronic metabolic disease characterized by persistent hyperglycemia caused by impaired insulin secretion or insulin resistance. Currently, the disease affects more than 415 million adults worldwide, and projections indicate that this number will rise to nearly 640 million by 2040 [1]. In addition to systemic complications such as cardiovascular disease, nephropathy, and neuropathy, diabetes also affects the eyes. Although diabetic retinopathy is the most recognizable symptom, diabetes also affects the anterior segment of the eye—including the cornea, conjunctiva, and lacrimal glands—contributing to ocular surface disease and dry eye syndrome [2]. The aim of this systematic review is to evaluate the potential of tear fluid biomarkers in diabetic ocular surface disease.

The tear film, composed of water, lipids, proteins, and mucins, plays a critical role in maintaining the integrity of the ocular surface. In diabetes, changes in tear composition may reflect both local ocular surface dysfunction and broader systemic changes [3]. Biomarkers, defined as objectively measurable indicators of biological or pathological processes, are increasingly being studied in tear fluid as potential diagnostic and monitoring tools. Tears are a particularly attractive medium for biomarker discovery: their collection is safe, non-invasive, and inexpensive, while reflecting both the health of the eyes and the entire body [4]. This makes tear-based biomarkers a promising method for the early detection and monitoring of diabetic complications of the ocular surface.

The tear film is increasingly recognized as a dynamic immunological and metabolic interface rather than a passive protective layer. As emphasized by Jiao et al. [5], its molecular composition reflects both local ocular surface homeostasis and systemic conditions such as diabetes, making it a valuable source of biomarkers for early disease detection.

Therefore, this systematic review summarizes current evidence on tear fluid biomarkers in diabetes, with a particular focus on their relevance for ocular surface disease.

## 2. Materials and Methods

This review was not registered in PROSPERO. We recognize that the lack of prospective registration and a predetermined protocol constitutes a methodological limitation, as it may increase the risk of selective reporting bias. To mitigate this risk, we strictly followed the PRISMA 2020 guidelines, predetermined the research question and eligibility criteria, and conducted independent and double selection, data extraction, and risk of bias assessment. Any discrepancies were resolved by consensus, and the complete search strategy, inclusion/exclusion criteria, and results of interest were fully reported to ensure transparency and reproducibility.

### 2.1. Eligibility Criteria

We included original studies published between January 2014 and June 2025 that evaluated tear fluid or ocular surface biomarkers in adult patients (≥18 years) with diabetes mellitus. Exclusion criteria were studies in children or adolescents, non-English publications, reviews, conference abstracts, and studies focusing on ocular surface biomarkers in diseases other than diabetes.

### 2.2. Search Strategy

A systematic search was performed in PubMed, Scopus, and Embase databases. The search terms were “biomarkers” AND (“ocular surface” OR “tear film” OR “conjunctiva”) AND (“diabetes” OR “diabetic”). Searches covered the period from January 2014 to June 2025, with the last search conducted in June 2025.

### 2.3. Study Selection and Data Extraction

Two reviewers (N.G. and A.M.) independently screened all titles and abstracts. Full texts were retrieved when eligibility could not be determined from the abstract. Disagreements were resolved through discussion or consultation with a third reviewer (A.N.). Reference lists of included articles were manually screened to identify additional relevant studies. Data were extracted independently by two reviewers using a standardized form, including study design, population characteristics, biomarker type, and main outcomes.

### 2.4. Risk of Bias Assessment

The methodological quality of included studies was assessed independently by three reviewers using the Joanna Briggs Institute (JBI) critical appraisal tools appropriate for each study design. Disagreements were resolved by consensus or, when necessary, by consulting a third reviewer. Overall, most studies were rated as having a moderate risk of bias, with frequent limitations such as small sample sizes, heterogeneous populations, and incomplete reporting of blinding or randomization. A detailed assessment using the JBI critical appraisal checklists for narrative reviews and original studies is provided in Appendix A.

After applying the inclusion and exclusion criteria and the final number of articles analyzed was 30. PRISMA flow diagram is illustrated in Figure 1.

## 3. Results

Of the 30 included studies, most were assessed as having a moderate risk of bias due to small sample sizes, heterogeneous populations, and variability in sample collection and analysis methods. Only a minority applied blinding or randomization procedures. These limitations should be taken into account when interpreting the findings.

### 3.1. Inflammatory Biomarkers

#### 3.1.1. Cytokines

Several studies have analyzed inflammatory cytokines in the tear fluid of patients with diabetes. IL-6 was evaluated in five studies, three of which showed significantly elevated levels in patients with diabetes [6,7,8], while two showed no significant differences [9,10]. In one study, IL-6 levels were significantly higher in proliferative retinopathy compared to non-proliferative retinopathy (47.7 pg/mL vs. 29.7 pg/mL, *p* < 0.001) [6]. Byambajav et al. [8] also confirmed elevated IL-6 levels in type 2 diabetes with accompanying dry eye syndrome compared to the control group, patients with dry eye syndrome alone, and diabetes without dry eye syndrome. Mechanistically, IL-6 contributes to ocular surface pathology by inducing acute phase proteins, activating MMP-9, stimulating Th17 differentiation, and promoting IL-1β secretion, leading to lacrimal gland cell apoptosis and corneal surface keratinization.

TNF-α was evaluated in 4 studies: 2 showed higher concentrations in diabetes [6,8], while 2 showed no significant changes [7,9]. One report showed significantly higher TNF-α levels in proliferative retinopathy compared to non-proliferative disease (13.5 pg/mL vs. 2.8 pg/mL) [6]. TNF-α is strongly associated with retinal cell death and blood-retinal barrier disruption, but it is also elevated in diabetic dry eye syndrome, limiting its diagnostic specificity.

IL-8 was examined in 3 studies, showing elevated levels in 2 [8,9] and no differences in 1 [7]. MCP-1 was evaluated in 2 studies, 1 of which showed elevated concentrations [7] and the other showed no significant changes [8]. Similarly, one study [7] found elevated IP-10 levels in patients with type 2 diabetes, but did not distinguish between stages of retinopathy. The inconsistency of the results is compounded by the frequent co-occurrence of dry eye syndrome, which in itself raises cytokine levels.

The involvement of additional mediators, such as IL-1ra, has also been suggested: Byambajav et al. [8] found lower levels in type 2 diabetes without DED compared to both diabetes with DED and the control group, suggesting a possible protective role. In patients with diabetic retinopathy, a shift in the Th1/Th2 cytokine balance toward Th1 has also been observed, reflecting a pro-inflammatory and cytotoxic environment [10].

Recently, MMP-9 has been investigated as a potential biomarker. Qu et al. [11] demonstrated significantly higher levels of MMP-9 in the tears of diabetic patients compared to the control group. In addition, Meera et al. [12] examined 31 inflammatory biomarkers in patients with diabetic peripheral neuropathy (DPN) and found that MMP-9 and TGF-β were elevated, although not statistically significant. Importantly, combining biomarkers in tears (MMP-9, TGF-β) with corneal esthesiometry improved diagnostic accuracy (AUC 84% compared to 65% for biomarkers alone), highlighting the potential value of multi-marker panels in combination with functional measurements.

In summary, these results indicate that IL-6 and IL-8 are the most commonly elevated cytokines in the tears of patients with diabetes, while TNF-α, MCP-1, and IP-10 show heterogeneous results across different studies. Their diagnostic specificity is limited because they are also elevated in non-diabetic ocular surface diseases, especially DED. Therefore, cytokine panels in combination with clinical or functional tests should be prioritized in the future, rather than relying on single markers.

#### 3.1.2. Neutrophil Extracellular Traps

Neutrophil extracellular traps (NETs) are part of the innate immune defense, but their excessive activation contributes to chronic inflammation and tissue damage. Elevated NET levels have been detected in the tear film of patients with type 2 diabetes and dry eye syndrome, where they correlated with increased inflammation and damage to the eye surface [13]. Their role in proliferative diabetic retinopathy is also suggested, as hyperglycemia and VEGF promote excessive NET formation, leading to microvascular dysfunction. Preclinical studies show that inhibition of NETs, for example, with DNase I, can promote corneal and retinal tissue regeneration, indicating that NETs may serve as both potential biomarkers and therapeutic targets in diabetic eye complications [14].

### 3.2. Emerging Biomarkers of Angiogenesis and Microvascular Dysfunction

#### 3.2.1. MicroRNA (miRNA)

High-throughput sequencing (HTS), also known as next-generation sequencing (NGS), enables direct sequencing of nucleic acids in clinical samples without the need for conventional methods [15]. This approach has significantly improved the understanding of disease mechanisms by providing genomic information and is increasingly applied to ocular surface research. In the context of diabetes, HTS has been used to characterize alterations in small RNAs, including microRNAs (miRNAs), which play important regulatory roles in inflammation, angiogenesis, and cellular apoptosis [16]. Several studies have shown that miRNA expression patterns are dysregulated in diabetes and may contribute to ocular surface complications. For example, changes in miRNA expression have been linked to pathways involved in cell signaling and epigenetic regulation [17]. These findings highlight the potential of HTS in identifying tear-based miRNA biomarkers that reflect early pathological processes in diabetes. Moreover, emerging evidence suggests that miRNAs may also interact with other regulatory molecules, such as long non-coding RNAs (lncRNAs), indicating a complex network of gene regulation in diabetic eye disease [18,19]. The identification of such molecules in tear fluid may provide novel diagnostic and prognostic biomarkers, as well as new therapeutic targets.

Studies have shown that the composition of microRNAs (miRNAs) in the tear film of patients with diabetic retinopathy (DR) is altered. For example, Sun et al. reported reduced expression of miR-23a in tears and serum of individuals with type 2 diabetes or prediabetes, suggesting its role as a potential inhibitor of VEGF and angiogenesis [20]. Hu et al. identified multiple miRNAs with altered expression in tears of DR patients. Among these, several (e.g., miR-9-5p, miR-143-3p, and miR-218-5p) were significantly upregulated, while others showed downregulation patterns. Importantly, miR-218-5p demonstrated a stepwise increase from healthy controls, through diabetic patients without retinopathy, to those with DR, highlighting its potential as a marker of disease progression [21].

#### 3.2.2. CircRNA

Circular RNAs (circRNAs) are a recently described subclass of non-coding RNAs formed through back-splicing, which gives them a stable, closed-loop structure. They regulate gene expression, modulate epigenetic mechanisms, and may influence cellular signaling pathways [22]. In diabetes-related eye disease, circRNAs have been associated with processes such as angiogenesis and microvascular dysfunction. Although most of the current data come from studies on the posterior segment, their biological stability and detectability in extracellular fluids suggest that circRNAs could also be measured in tears. This opens the possibility of using circRNAs as non-invasive biomarkers to reflect early ocular surface involvement in diabetes. Their integration into tear-based biomarker panels may complement other RNA molecules, such as microRNAs, to improve early diagnosis and disease monitoring [23].

#### 3.2.3. Vascular Endothelial Growth Factor (VEGF)

Another example is vascular endothelial growth factor (VEGF), which is involved in blood vessel formation in many diseases and has been linked to the development of diabetic retinopathy. VEGF levels in the vitreous body and the aqueous fluid increase during the active proliferative phase of diabetic retinopathy and decrease after treatment, and similar changes are also observed in the tears. However, VEGF can enter tears not only from the retina but also from the corneal stroma and conjunctiva [24]. In combination with lipocalin 1 (LCN1), it appears to be a useful biomarker for screening, achieving an accuracy of more than 80% [25].

In a pilot study by Štorm et al. [26], which assessed ocular surface biomarkers in type 1 diabetes (T1D), no significant differences were observed in tear MMP-9 levels, HLA-DR expression, or tear osmolarity between T1D patients and healthy controls. The only parameter that distinguished the groups was a marked reduction in corneal nerve fiber length (CNFL) in patients with T1D. These findings indicate that conventional tear biomarkers, such as MMP-9 and HLA-DR, may not be sufficiently sensitive in the early stages of T1D, whereas corneal neuropathy, reflected by decreased CNFL, could serve as a more reliable early indicator of ocular surface involvement.

Qu et al. [27] conducted a clinical study to assess tear MMP-9 concentrations and their relationship with ocular surface parameters in diabetes. Two groups were compared: patients with diabetic dry eye (DED) and patients with diabetes without DED. Tear MMP-9 levels were significantly higher in the DED group and correlated with reduced tear film stability, decreased corneal sensitivity, and greater severity of subjective dry eye symptoms. The authors concluded that tear MMP-9 may serve as a valuable diagnostic and monitoring biomarker in diabetic DED and could support patient selection for anti-inflammatory therapies.

Evidence regarding angiogenesis-related markers such as VEGF, miRNA, and circRNA comes mainly from small pilot studies with limited sample sizes [2,6,17,20,21,22] These exploratory studies suggest a potential role for these molecules in the development of diabetic microvascular complications; however, the current findings should be considered preliminary and require confirmation in larger, well-designed cohorts.

### 3.3. Tear Film Composition and Other Biomarkers

#### 3.3.1. Glucose

There have been multiple attempts to measure glucose levels in the tear film of patients with diabetes. Although several studies reported higher glucose concentrations in tears compared with healthy individuals, the results remain inconsistent. In one study, both tear and serum glucose increased after an oral glucose load; however, no clear correlation was observed between them in either healthy or diabetic subjects. The lack of consistent association may be related to methodological variability, individual patient characteristics, or contamination of tear fluid with blood traces. Taken together, these findings suggest that while tear glucose may reflect systemic metabolic alterations, its clinical utility as a biomarker of diabetes remains uncertain [24].

#### 3.3.2. Meibum Lipids

The lipid layer of the tear film originates from meibum secreted by the Meibomian glands and plays a key role in maintaining tear film stability. Changes in the composition of meibum can weaken its protective and moisturizing functions, thereby contributing to ocular surface dysfunction. In a case–control study involving 30 patients with type 2 diabetes and dry eye syndrome compared to 30 healthy control subjects, Yang et al. analyzed the composition of meibum using high-performance liquid chromatography coupled with mass spectrometry (HPLC-MS). They noted a significant reduction in triglyceride and oleic hydroxy fatty acid (OAHFA) concentrations, accompanied by an increase in cholesterol esters, phospholipids, sphingomyelin, and glucosylceramide concentrations. These changes were correlated with shorter tear film breakup time (TBUT), increased tear osmolarity, and higher scores on the Ocular Surface Disease Index (OSDI) scale. These results suggest that the change in the lipid composition of meibum in diabetes is not only a biochemical alteration but also has clinical significance, as it contributes to tear film instability and the onset of dry eye syndrome symptoms [28].

In addition, glycomic analyses of tears have provided preliminary insights into diabetes-related alterations in tear film composition [27]. However, the current evidence on lipid and glycomic markers remains very limited and is derived from small-scale studies. Therefore, these findings should be regarded as exploratory, and lipidomic alterations in diabetes should be considered emerging biomarkers that require replication and validation in larger, independent cohorts before their clinical utility can be established.

#### 3.3.3. Metallic Elements

Studies on tear film composition suggest that trace elements may contribute to the development of type 2 diabetes and its ocular complications. Cancarini et al. compared metal levels in the tears and serum of patients with diabetes and healthy controls, identifying significant differences in several elements, including chromium, zinc, cobalt, manganese, lead, and barium as shown in Table 1. These alterations may be linked to impaired microcirculation, changes in vascular permeability, and disruption of the blood–tear barrier, which could exacerbate inflammatory processes and ocular surface degeneration. While these findings indicate that tear trace element analysis may provide insight into metabolic disturbances in diabetes, their diagnostic and prognostic relevance for ocular surface disease remains uncertain and requires further investigation [29,30].

### 3.4. Epithelial and Mucin-Related Changes

#### Proteins

Advanced glycation end products (AGEs) are consistently elevated in patients with diabetes and correlate with HbA1c, reflecting chronic hyperglycemia. However, their presence in some healthy individuals reduces their diagnostic specificity [24]. In addition to AGEs, proteomic analyses of tears have revealed alterations in several structural and stress-related proteins. Elevated levels of lipocalin-1, extracellular matrix-binding proteins, calcium-binding proteins S100A8 and S100A9, keratin 4, heat shock proteins 70 (HSP70), and Ig lambda chain suggest an ongoing inflammatory and immune response. On the other hand, beta-2 microglobulin (B2M), typically expressed on the surface of most cells, was reduced in diabetic tears [31].

Other studies have identified additional proteins of interest, including lactoferrin, lactatin, lysozyme C, and lipophilin A, which may be particularly relevant in the context of diabetic retinopathy, where they appear to reflect vascular changes and proliferative activity [24].

Insulin-like growth factor-binding protein 3 (IGFBP-3) has emerged as one of the most promising tear-based biomarkers associated with diabetic corneal neuropathy. Its role is particularly important because it regulates the bioavailability of IGF-1, a neuroprotective factor important for corneal nerve integrity. In diabetes, a shift toward IGFBP-3 dominance reduces IGF-1 activity, thereby promoting apoptosis and nerve fiber degeneration [1]. The study by Stuard et al. demonstrated that IGFBP-3 concentrations were significantly higher in the tears of patients with diabetes compared to healthy controls. Importantly, increased IGFBP-3 levels correlated with reduced corneal nerve fiber length and branching density, suggesting that overexpression of this protein may serve as an early marker of neuropathy in type 2 diabetes [9,32].

It should be noted that many of these proteins may also be altered in dry eye disease (DED), a common comorbidity of diabetes, complicating their interpretation and highlighting the need for multi-marker panels and validation in larger cohorts. The main proteomic changes in the tears of diabetic patients are summarized in Table 2.

Proteomic studies have identified several potential protein biomarkers (Table 2). Among these, IGFBP-3 has shown the most consistent associations across multiple independent studies [14,32], confirming its potential as a reliable biomarker of corneal nerve changes in diabetes. In contrast, other proteins, such as inflammatory cytokines, lipocalin-1, or MMP-9, have yielded variable results, with some studies showing significant changes and others showing no clear differences [6,9,11,33]. Overall, IGFBP-3 appears to be the most reliable potential protein marker, while other proteins should be considered preliminary findings.

### 3.5. Neurodegeneration-Related Markers

#### 3.5.1. Substance P and Neuropeptide Y

Peripheral nerve damage is one of the most common complications of diabetes, and early detection can prevent serious consequences such as foot ulcers and amputations. Since corneal nerves undergo morphological changes even before clinical symptoms appear, increasing attention is being paid to neuropeptides in the tear film as potential biomarkers of this process [34,35].

Studies have shown that substance P (SP) is present in lower concentrations in the tears of people with diabetes. Its level correlates with the degree of corneal nerve fiber damage and the severity of neuropathy. This association has been reported in both type 1 and type 2 diabetes, although the strongest and most consistent associations have been demonstrated in type 2 diabetes [35]. Some studies have reported a weaker or less consistent association in type 1 diabetes, which may limit the diagnostic significance of SP in this group [2,24,36]. Despite these differences, the overall evidence suggests that a reduction in SP reflects diabetes-related corneal neurodegeneration.

Neuropeptide Y (NPY) has been proposed as another potential biomarker of microangiopathy in type 1 diabetes. Studies have shown that NPY levels are reduced in patients with peripheral neuropathy and early diabetic retinopathy. Lower NPY concentrations correlated with adverse morphological changes in corneal nerves, such as shorter and sparser fibers, suggesting its role in detecting early microvascular complications [14,37]. However, these results are based on relatively small study groups, and further research is needed to confirm its clinical utility.

In summary, SP and NPY appear to provide complementary rather than conflicting information. SP is a more established biomarker, supported by numerous studies and particularly valuable in type 2 diabetes, while NPY is a new candidate with preliminary evidence in type 1 diabetes. Both highlight diabetes-related corneal neurodegeneration, but larger comparative studies are needed to confirm their diagnostic value in different types of diabetes.

#### 3.5.2. Nerve Density and Length

It has been shown that reduced corneal nerve density correlates negatively with the severity of peripheral neuropathy and diabetic retinopathy. Nerve morphology was assessed using in vivo confocal microscopy (IVCM), a non-invasive imaging technique that allows high-resolution visualization of the subbasal corneal nerve plexus. For the diagnosis of diabetic neuropathy, the sensitivity and specificity of nerve density were 82% and 52%, respectively. Importantly, reduced corneal nerve density may precede the development of peripheral complications of diabetes. A negative correlation between mean nerve density and HbA1c levels was also observed, and nerve density improved after achieving better glycemic control [38]. Although IVCM provides valuable diagnostic information, its use in routine clinical practice is currently limited to specialized centers due to the availability of equipment and the need for qualified personnel.

#### 3.5.3. Progranulin

Progranulin (PGRN) is a glycoprotein with neuroprotective and anti-inflammatory properties, present in several body fluids, including the tear film [39]. It is suggested that it plays an important role in maintaining corneal nerve integrity and eye surface homeostasis [40]. Zhou et al. examined PGRN levels in the tears of patients with type 2 diabetes and compared them with those in healthy control subjects. They found significantly reduced PGRN concentrations in diabetic patients, with the lowest levels observed in those with concomitant diabetic retinopathy. Importantly, PGRN levels were positively correlated with corneal nerve parameters such as corneal nerve fiber density (CNFD), length (CNFL), and branch density (CNBD). In addition, reduced PGRN levels were associated with more severe symptoms of dry eye syndrome, including shorter tear break-up time (TBUT) and reduced Schirmer test scores [41]. These findings indicate that low PGRN levels in tears may reflect progressive corneal neuropathy and ocular surface dysfunction in diabetes, suggesting its potential as a non-invasive biomarker. However, the study was limited by its relatively small sample size, and further well-designed studies are needed to confirm the diagnostic value of PGRN and to clarify whether it may also serve as a therapeutic target in diabetic eye complications.

### 3.6. Oxidative Stress and Metabolic Stress Markers

#### 3.6.1. Glycosaminoglycans (GAGs)

Glycosaminoglycans (GAGs) are primarily composed of disaccharide units that form polysaccharide chains. They perform various functions throughout the body, including serving as components of the extracellular matrix and connective tissue, participating in cell signaling pathways, and playing a role in wound healing. Their excretion in urine has been proposed as a marker for both type 1 and type 2 diabetes. It appears that they may function similarly in the tear film [24]. Increased GAG levels have been detected in individuals with diabetes, with type 2 diabetes producing a stronger response than type 1. In cases of proliferative retinopathy, GAG levels were higher compared to the non-proliferative form.

#### 3.6.2. Glycans

A study conducted by Nguyen-Khuong et al. [27] demonstrated that, despite the overall stability of the glycan profile in tears, changes occur in the low-abundance glycan structures, which may be associated with diabetes and its complications. Using mass spectrometry analysis (LC-ESI-IT-MS/MS), 50 N-linked glycans and 8 O-linked glycans were identified, with changes in abundance observed only for glycans with a low content (<5%). Significant quantitative changes in the abundance of selected glycans are presented in Table 3.

The early increase in sialylated hybrid glycan and monosialylated biantennary glycan with H-type 2 antigen in diabetes may be related to the body’s compensatory mechanisms, while their decrease in DR suggests involvement in disease progression. Decreased levels of monosialylated hybrid glycan in DR suggest that it may be important for maintaining ocular surface homeostasis in diabetes. Increased bisection of GlcNAc biantennary glycan in DR indicates that it may play a role in inflammatory processes and disease progression. Decreased levels of biantennary monosialylated glycan with H-type 2 antigen may be an early indicator of glycosylation damage associated with diabetes. With regard to O-glycans, statistically significant changes were shown only for disialylated core glycan 2, whose increase in DR may signify an enhanced immune or inflammatory response [42].

A study conducted by Nguyen-Khuong et al. showed that the glycan profile in tears reflects both metabolic and disease-related changes. Changes in low-content N- and O-glycans were associated with diabetes and diabetic retinopathy, suggesting their potential as early diagnostic indicators. Unlike HbA1c, which provides systemic information on glycemic control, tear glycans can record local inflammatory and metabolic processes on the surface of the eye. Although research is still in the exploratory phase, these findings open up new diagnostic possibilities for both type 1 and type 2 diabetes [6].

#### 3.6.3. Glycated Albumin (GA)

Glycated albumin has recently gained attention as a potential alternative to HbA1c in assessing glycemic control. In diabetes, albumin is more susceptible to glycation at multiple lysine residues under hyperglycemic conditions, and elevated levels have also been detected in tears. Tan et al. demonstrated a strong correlation between GA levels in tears and plasma (R = 0.92, *p* < 0.01), with a diagnostic accuracy of 0.97, which was even higher than in plasma (AUC 0.80) [31]. Importantly, GA reflects the state of glycemia over the past 2–3 weeks, similar to HbA1c, but is less subject to short-term fluctuations in diet or blood glucose levels and is less susceptible to sampling artifacts than glucose in tears. These properties make GA a particularly attractive tear-based biomarker in both type 1 and type 2 diabetes. Its stability and non-invasive availability suggest potential applications not only in diagnosis but also in long-term monitoring of metabolic control and ophthalmic complications. However, further validation in larger and more diverse cohorts is necessary before GA can be recommended as a routine clinical tool [43].

#### 3.6.4. Oxidative Stress (OS)

Oxidative stress plays a key role in the pathogenesis of DR, leading to damage to retinal cells and blood vessels. The high metabolic activity of the retina, along with exposure to light and UV radiation, promotes the production of reactive oxygen species (ROS). Combined with chronic hyperglycemia, this leads to endothelial dysfunction of retinal vessels and activation of pro-inflammatory pathways and lipid peroxidation. Studies have shown that oxidation–reduction imbalance occurs in DR. Table 4 shows the most important antioxidant enzymes along with the change in their concentration levels compared to healthy subjects [6].

Oxidative stress may be one of the main therapeutic targets in the prevention and treatment of DR, as it is a process that leads to vascular dysfunction and neurodegeneration. Further research on OS biomarkers may prove useful in developing earlier diagnostic methods for DR [6].

## 4. Discussion

This systematic review summarizes the current state of knowledge on tear fluid biomarkers in diabetes, focusing on proteins, lipids, neuropeptides, cytokines, and nucleic acids. Using the PRISMA guidelines, 30 studies were identified and summarized that analyzed a wide range of biomarker classes in relation to diabetes complications. A summary of key results and biomarker categories is presented in the Table 5.

Compared to previous narrative reviews, our findings provide several important insights. Hagan (2016) highlighted the potential of tear biomarkers in systemic diseases but mainly discussed exploratory study results without a systematic synthesis [3]. Markoulli (2017) reviewed corneal nerve morphology and tear film changes in diabetes but focused mainly on structural and imaging markers [35]. Winiarczyk (2022) presented an overview of proteomic changes in eye diseases but did not comprehensively integrate data on multiple types of biomarkers [10]. In contrast, this review systematically integrates protein, lipid, neuropeptide, and nucleic acid markers, emphasizing their complementary role in reflecting different pathophysiological aspects of diabetes.

The main advantage of this review is its systematic methodology, compliance with the PRISMA 2020 guidelines, and broad coverage of different classes of biomarkers. By combining multiple categories of biomarkers, our review provides a more holistic overview of tear-based diabetes diagnostics than previous narrative reviews. In addition, the tabular presentation of results facilitates a clear comparison of the effectiveness of biomarkers across different studies.

However, several limitations should be noted. The review was not registered in PROSPERO, and no predefined protocol was prepared, which may increase the risk of selective reporting bias. However, we mitigated this by strictly adhering to the PRISMA guidelines and transparently reporting all stages of study selection and data extraction. The included studies were heterogeneous in terms of design, patient populations, and analytical methods, which prevented meta-analysis. In addition, many studies involved small samples or pilot studies, which limited the generalizability of the results.

Future studies should aim to validate the most promising biomarkers, such as IGFBP-3 and substance P, in larger and well-characterized cohorts. The development of standard protocols for tear collection and analysis is essential to improve comparability between studies. Multimarker panels combining proteins, lipids, and neuropeptides may increase diagnostic accuracy by capturing the multifactorial nature of diabetic eye and systemic complications. Furthermore, advances in biosensor technology may facilitate the translation of tear biomarkers into point-of-care diagnostics, enabling noninvasive real-time monitoring of disease progression.

In conclusion, this systematic review highlights both the promising prospects and current limitations of tear biomarkers in diabetes. Although several candidates show potential, particularly IGFBP-3 and substance P, further validation and methodological standardization are necessary before tear biomarkers can be incorporated into clinical practice.

## 5. Conclusions

This systematic review highlights the growing body of evidence indicating that tear fluid contains multiple classes of biomarkers reflecting both ocular and systemic complications of diabetes. Among the molecules studied, the strongest and most consistent evidence supports the role of IGFBP-3, progranulin, and changes in glycan signatures as promising candidates for detecting diabetes-related corneal neurodegeneration and metabolic disorders. In contrast, commonly studied inflammatory cytokines such as IL-6, TNF-α, and the angiogenic factor VEGF, although often found in elevated concentrations, are not sufficiently specific due to overlap with other ocular surface conditions and therefore have limited diagnostic value as stand-alone indicators.

The most realistic clinical applications of tear biomarkers in the near future are for non-invasive screening, disease progression monitoring, and risk stratification of diabetic complications. Their use in multi-marker panels, integrated with clinical parameters, may increase diagnostic accuracy and overcome the limitations of single markers. Standardization of tear collection and analysis protocols, validation in larger cohorts, and the development of biosensor-based detection platforms will be essential steps toward clinical application.

## Figures and Tables

**Figure 1 jcm-14-06958-f001:**
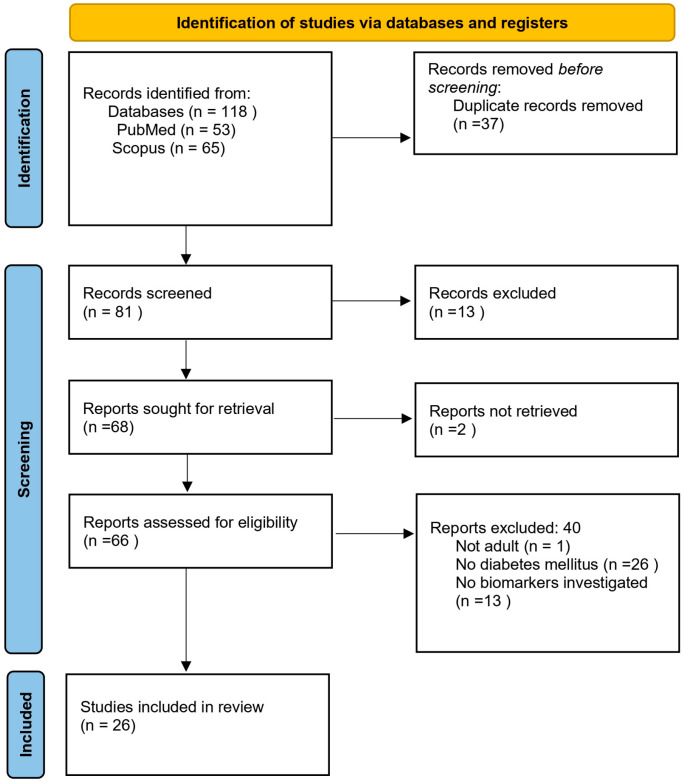
Preferred Reporting Items for Systematic Reviews and Meta-Analyses (PRISMA) flow diagram.

**Table 1 jcm-14-06958-t001:** Comparison of trace metal concentrations in the tear film of patients with type 2 diabetes and healthy individuals [29].

Element Name	Patients with DM [ng/mL]	Healthy Individuals [ng/mL]	Statistical Significance
Zinc (Zn)	66.00 (39.50–83.00)	33.25 (23.20–52.50)	*p* = 0.0002
Chromium (Cr)	0.75 (0.35–1.10)	0.25 (0.15–0.40)	*p* < 0.0001
Cobalt (Co)	8.22 (2.12–11.75)	1.77 (0.04–3.40)	*p* < 0.0001
Manganese (Mn)	7.65 (4.70–10.30)	3.87 (2.25–5.95)	*p* < 0.0001
Barium (Ba)	5.35 (3.60–6.80)	2.60 (1.45–4.00)	*p* < 0.0001
Lead (Pb)	1.15 (0.70–1.40)	0.50 (0.09–0.90)	*p* < 0.0001

**Table 2 jcm-14-06958-t002:** Key proteomic biomarkers in tears of diabetic patients.

Protein/Group	Change in Diabetes	Clinical Significance	Comments/Limitations
AGEs	Increased (correlation with HbA1c)	Reflect chronic hyperglycaemia	Also present in healthy individuals → low specificity
Lipocalin-1	Increased	Inflammation, potential marker	Also influenced by DED
Proteins binding the extracellular space	Increased	Tissue remodeling, inflammatory processes	Limited validation
S100A8/S100A9	Increased	Calcium-binding proteins; inflammatory processes	Non-specific, also elevated in other eye diseases
Keratin 4	Increased	Marker of epithelial stress	Requires confirmation
Heat shock protein 70 (HSP70)	Increased	Response to cellular stress	Non-specific, also affected by DED
Beta-2 microglobulin (B2M)	Decreased	Potential tumour/immunological marker	Limited diagnostic value in DM
Immunoglobulin lambda chain	Increased	Immune/inflammatory activity	May reflect DR progression, also influenced by DED
Lactoferrin, Lactatin, Lysozyme C, Lipophilin A	Increased	Inflammatory processes, changes associated with DR	Small study groups; influence of DED
IGFBP-3	Increased (3-fold in DM)	Strong correlation with reduced corneal nerve fibre density; marker of neuropathy	Promising but limited data; independent of HbA1c
IGF-1	Increased activity (bound by IGFBP-3)	Physiologically neuroprotective	Impaired balance promotes nerve damage

**Table 3 jcm-14-06958-t003:** Comparison of selected glycan levels in the tear film of healthy individuals (CON), patients with diabetes without retinopathy (DM), and patients with diabetic retinopathy (DR). Results are expressed as the mean percentage composition of the given glycan along with the standard deviation. Data adapted from [6].

Glycan	Type	CON [%]	DM [%]	DR [%]	*p*-Value
Sialylated hybrid glycan	N-glycan	0.05 ± 0.0	0.3 ± 0.3	0.0005 ± 0.0	*p* < 0.01
Monosialylated biantennary glycan with H-type 2 antigen	N-glycan	0.05 ± 0.0	0.36 ± 0.2	0.05 ± 0	*p* < 0.05
Monosialylated hybrid glycan	N-glycan	0.19 ± 0.1	0.46 ± 0.1	0.05 ± 0.0	*p* < 0.05
Bisecting GlcNAc biantennary glycan	N-glycan	0.47 ± 0.0	0.46 ± 0.1	1.1 ± 0.3	*p* < 0.001
Biantennary monosialylated glycan with H-type 2 antigen	N-glycan	1.1 ± 0.6	0.05 ± 0.0	0.05 ± 0.0	*p* < 0.01
Disialylated core 2 glycan	O-glycan	0.4 ± 0.2	0.8 ± 0.5	3.9 ± 1.9	*p* < 0.05

**Table 4 jcm-14-06958-t004:** Changes in the activity of antioxidant enzymes and levels of oxidative stress biomarkers in the tear film of patients with diabetic retinopathy (DR), source [6].

Enzyme Name	Biological Significance	Level in DR Patients (Compared to Healthy Individuals)
Superoxide dismutase (SOD)	Enzyme neutralizing oxygen radicals	Decreased
Glutathione (GSH)	Antioxidant	Decreased
Glutathione peroxidase (GPx)	Enzyme reducing peroxides	Decreased
Catalase (CAT)	Enzyme detoxifying hydrogen peroxide	Decreased
Malondialdehyde (MDA)	Lipid peroxidation marker	Increased

Statistical significance values (*p*) were not provided in the original study.

**Table 5 jcm-14-06958-t005:** Biomarkers in ocular surface diabetic eye disease.

First Author	Year	Study Design	Origin of Study	Population	Main Findings	Funding Sources	Conflicts of Interest	Biomarkers Investigated
He et al. [15]	2022	Narrative Review	Capital Medical University, China	Not applicable	HTS has identified specific ocular surface microbiota and non-coding RNAs such as miRNAs and lncRNAs in diabetic patients, supporting their role in early diagnosis of diabetic ocular surface disorders.	Beijing Natural Science Foundation and National Natural Science Foundation of China	None declared	miRNAs, lncRNAs,
Alotaibi et al. [24]	2022	Narrative Review	University of New South Wales, Australia	Not applicable	Tear biomarkers, including AGE, S100A8/A9, lipocalin-1, IGFBP-3, and substance P, are associated with ocular surface inflammation, nerve damage, and diabetes-related complications.	Not specified	None declared	AGE, S100A8, S100A9, lipocalin-1, IGFBP-3, Substance P, IP-10, B2M, HSP70, keratin 4, miRNAs
Ting et al. [31]	2016	Narrative Review	Singapore National Eye Centre and Singapore Eye Research Institute, Singapore	Not applicable	Cytokines and neuropeptides in the tear film are highlighted as non-invasive biomarkers for DR diagnosis and progression monitoring.	Not specified	Personal fees from Abbott, Novartis, Pfizer, Allergan, and Bayer (Tien Yin Wong)	Cytokines, neuropeptides
Winiarczyk et al. [10]	2022	Narrative Review	Medical University of Lublin, Poland	Not applicable	Lipocalin-1 (LCN1), VEGF, and MMP-9 in tears demonstrate diagnostic potential for both DR and DED.	National Science Centre, Poland (grant no. 2017/25/N/NZ5/01875)	None declared	LCN1, VEGF, MMP-9
Zhang et al. [22]	2020	Narrative Review	Eye Center of the Second Affiliated Hospital, Zhejiang University, China	Not applicable	Circular RNAs such as circ_0005015 and circZNF609 are upregulated in diabetic retinopathy and may regulate angiogenesis, offering new diagnostic targets.	Not specified	None declared	circ_0005015, circZNF609
Toh et al. [34]	2024	Narrative Review	Lee Kong Chian School of Medicine, Nanyang Technological University, Singapore	Not applicable	Corneal neuromas are observed in patients with ocular surface diseases, including diabetic neuropathy. Their presence is associated with altered corneal sensitivity, pain, and dryness. Neuromas may serve as potential indicators of nerve regeneration and biomarkers of corneal nerve damage in diabetes.	Not specified	None declared	CircRNAs (e.g., circHIPK3)
Markoulli et al. [35]	2017	Cross-sectional	School of Optometry and Vision Science, University of New South Wales, Australia	9 with diabetes, 17 healthy controls	Tear film substance P was significantly lower in people with diabetes and correlated positively with corneal nerve fiber density. Substance P may serve as a non-invasive biomarker of corneal nerve health in diabetic neuropathy.	UNSW startup funds; ARC Future Fellowship	None declared	Neutrophil extracellular traps (neutrophil extracellular traps (NETs))
Ma et al. [39]	2021	Narrative Review	School of Optometry, The Hong Kong Polytechnic University, Hong Kong SAR, China	Not applicable	Mass spectrometry–based proteomics enables precise identification of tear film biomarkers associated with ocular surface changes in multifactorial diseases. Specific proteins such as lactoferrin, lipocalin-1, lysozyme, and S100 family members show altered expression in conditions like diabetic eye disease and dry eye syndrome, highlighting their potential utility as non-invasive diagnostic and prognostic biomarkers.	Not specified	None declared	lactoferrin, lysozyme, S100 family
Lagali et al. [14]	2018	Clinical observational	Institute for Clinical and Experimental Medicine, Linköping University, Sweden	81 participants: 39 with DM2, 42 controls (33 NGT, 9 IGT); stratified by diabetes duration	In early type 2 diabetes, increased density and maturation of dendritic cells are observed in the corneal epithelium. These changes are associated with upregulated TNFRSF9 expression, suggesting subclinical immune activation and early inflammatory remodeling of the diabetic ocular surface.	Not specified	None declared	TNFRSF9, mature and immature dendritic cells
Mansoor et al. [1]	2020	Narrative Review	Singapore Eye Research Institute and Duke-NUS Medical School, Singapore	Not applicable	Corneal nerve loss in diabetes leads to epithelial damage and tear film instability. IVCM parameters and tear biomarkers like IGFB-3 and Substance P serve as early, non-invasive indicators of diabetic corneal neuropathy.	Not specified	None declared	IGFBP-3, Substance P
Ozdalgic et al. [12]	2022	Narrative Review	Koç University Translational Medicine Research Center and Department of Mechanical Engineering, Koç University, Turkey	Not applicable	EIS is a novel approach for detecting tear-based diabetic biomarkers, including glucose, lactate, and TNF-α. It offers real-time, non-invasive monitoring of metabolic and inflammatory alterations on the ocular surface in diabetes.	TUBITAK, Alexander von Humboldt Foundation, Marie Skłodowska-Curie Fellowship, Royal Academy Newton-Katip Çelebi Partnership	None declared	Glucose, lactate, TNF-α (via electrochemical impedance spectroscopy)
Stuard et al. [44]	2020	Narrative Review	University of Texas Southwestern Medical Center, USA	Not applicable	IGFBP-3 levels in tears are elevated in type 2 diabetes and correlate with corneal nerve fiber degeneration, independent of HbA1c levels.	National Eye Institute (NIH)	None declared	IGFBP-3
Nguyen-Khuong et al. [27]	2015	Experimental	Biomolecular Research Institute, Macquarie University, Australia	12 healthy, 8 with diabetes without DR, 11 with DR	Detailed glycomic profiling of basal tears via LC-MS/MS showed a high interindividual conservation of tear glycan structures. However, five low-abundance N-linked glycans and one disialylated core 2 O-glycan showed significant alterations in patients with diabetes and DR, suggesting their potential as early biomarkers.	Australian Research Council (ARC Discovery Grant DP1094624), APAF (NCRIS program)	None declared	Specific N-glycans (e.g., bisecting GlcNAc, sialylated hybrid, fucosylated biantennary) and one disialylated core 2 O-glycan in tears
López-Contreras et al. [6]	2020	Narrative Review	University of Guadalajara, Mexico	Not applicable	Elevated oxidative stress markers (e.g., MDA) and inflammatory cytokines (IL-6, TNF-α, VEGF) in tears are linked to diabetic retinopathy severity.	Not specified	None declared	MDA, SOD, GPx, IL-6, TNF-α, VEGF
Han et al. [2]	2019	Narrative Review	South Korea	Not applicable	Diabetes affects the anterior segment of the eye, leading to corneal neuropathy, epithelial damage, and dry eye. Corneal nerve changes may precede other complications, and corneal confocal microscopy can serve as an early biomarker. Treatments include insulin, IGF-1, NGF, naltrexone, and antioxidants.	Grant from Kangwon National University (2018)	None declared	Corneal nerve parameters (density, length, branching), substance P, NGF, IGFBP-3
Tan et al. [43]	2024	Experimental, comparative	University of Illinois, USA	8 patients with diabetes (DM1/DM2), 8 healthy controls	Glycated albumin levels in tears show strong correlation with plasma levels and serve as a stable, non-invasive biomarker of glycemic control.	American Diabetes Association grant no. 1-18-VSN-19; NIH (NIDA) grant no. P30DA018310	Two authors are founders of InnSight Tech, which had no financial involvement in the study.	Glycated albumin
Altman et al. [21]	2023	Narrative Review	Augusta University, USA	Not applicable	Tear microRNAs are stable, non-invasive biomarkers detectable in the tear film. Several miRNAs were identified as being dysregulated in ocular diseases like diabetic retinopathy. MiRNAs in tears show promise for diagnostics and monitoring disease progression.	NIH Grants R01 EY029728, R01 EY026936, P30 EY031631	None declared	Tear miRNAs
Tummanapalli et al. [36]	2019	Prospective cross-sectional	University of New South Wales & Prince of Wales Hospital, Australia	63 patients with DM1 or DM2, 34 healthy controls	Tear film substance P was significantly lower in DM1 patients with peripheral neuropathy and correlated with nerve parameters—potential biomarker for Diabetic Peripheral Neuropathy.	Not reported	None declared	Substance P
Byambajav et al. [8]	2023	Observational comparative	Glasgow Caledonian University, UK	47 with DM2 + DED, 41 with DM2, 17 with DED-only, 17 healthy controls	Tear IL-6 and IL-8 levels were significantly elevated in DM2 patients with DED and correlated with clinical symptoms—potential biomarkers for DM2-DED.	Ph.D. Research Studentship, School of Health and Life Sciences, GCU	None declared	IL-6, IL-8
Hagan et al. [3]	2016	Narrative Review	Glasgow Caledonian University, UK	Not applicable	Tear proteins such as lactotransferrin, lysozyme C, lipocalin-1, β2-microglobulin, NGF, and TNF-α are reported as altered in diabetic patients and proposed as noninvasive indicators of early retinal and corneal damage.	Not reported	None declared	LCN-1, lactotransferrin, lysozyme C, lacritin, B2M, HSP27, TNF-α, NGF
Zhou et al. [41]	2024	Prospective, cross-sectional	Shanghai Ninth People’s Hospital, China	48 patients with DR (DM2), 22 healthy controls	Progranulin levels are significantly lower in diabetic patients and correlate with corneal nerve loss and dry eye parameters.	National Natural Science Foundation of China (82271041, 82070919), SJTU Research Programs	None declared	Progranulin
Liu et al. [9]	2019	Cross-sectional comparative	Peking University Third Hospital, Beijing, China	32 DM2 with DED, 24 DM2 without DED, 28 non-diabetic DED, 29 healthy controls	Tear EGF levels were significantly elevated in patients with DM2 and DED. No significant differences in IL-17A, IL-1β, and TNF-α were found in diabetes-related DED vs. controls, while these cytokines were elevated only in non-diabetic DED, suggesting differing inflammatory mechanisms.	National Natural Science Foundation of China, Scientific Research Foundation for Returned Overseas Chinese Scholars	None declared	EGF, IL-1β, IL-17A, TNF-α
Stuard et al. [32]	2017	Prospective cross-sectional	University of Texas Southwestern Medical Center, USA	12 patients with DM2, 8 healthy controls	Elevated tear IGFBP-3 levels are strongly associated with reduced corneal nerve length and branching in type 2 diabetes, serving as an early neuropathy indicator.	NIH/National Eye Institute: R21 EY024433, R01 EY024546 Core Grant: P30 EY020799 Unrestricted grant from Research to Prevent Blindness (New York, NY, USA)	None declared	IGFBP-3
Britten-Jones et al. [37]	2024	Cross-sectional	The University of Melbourne & St Vincent’s Hospital, Melbourne, Australia	41 patients with DM1 (with and without DR and SFN), 22 healthy controls	Tear neuropeptide Y (NPY) levels are reduced in type 1 diabetes patients with early-stage retinopathy and neuropathy, indicating microvascular impairment.	Rebecca L. Cooper Medical Research Foundation grant; Australian Government Research Training Program (ACBJ); University of Melbourne Postdoctoral Fellowship	Some authors hold a provisional patent for tear NPY as a biomarker; others declared no conflict.	Neuropeptide Y
Li et al. [13]	2022	Narrative Review	Department of Ophthalmology, Peking University First Hospital, Beijing, China	Not applicable	Neutrophil extracellular traps serve both protective and pathological roles in the eye. While they defend against pathogens, excessive NET formation may promote inflammation, thrombosis, and autoimmunity. Neutrophil extracellular traps show potential as biomarkers and therapeutic targets.	National Natural Science Foundation of China	None declared	NET
Cancarini et al. [29]	2017	Cross-sectional observational	University of Brescia, Italy	47 patients with DM2 (all with DR), 50 non-diabetic controls with other ocular diseases	Patients with DM2 and DR showed significantly elevated tear concentrations of Zn, Cr, Co, Mn, Ba, and Pb compared to controls. Trace elements in the tear film, particularly chromium and cobalt, may serve as potential biomarkers of diabetes-related microangiopathy.	University of Brescia	None declared	Zn, Cr, Co, Mn, Ba, Pb
Štorm et al. [26]	2025	Pilot cross-sectional study	Charles University, Czech Republic	19 patients with DM1 (various DR stages) vs. 15 healthy controls	Significant reduction in corneal nerve fiber length (CNFL) in DM1 patients. No significant differences in tear osmolarity, TBUT, Oxford score, MMP-9 levels, or HLA-DR expression.	Not reported	None declared	MMP-9, HLA-DR, tear osmolarity, CNFL.
Qu et al. [11]	2025	Cross-sectional study	He Eye Specialist Hospital, Shenyang, China	144 patients with DM2 screened, 110 included (55 with DED], 55 with [DNDE])	Tear MMP-9 concentrations were significantly higher in DDE compared with DNDE; higher MMP-9 correlated with shorter NITBUT, lower corneal sensitivity,	Not reported	None declared	MMP-9
Qin et al. [33]	2025	Cross-sectiona	University of Houston + China collaboration	Healthy controls, patients with pre-diabetes, patients with DM2 without retinopathy	Identified a panel of 17 proteins with differential expression across groups (notably cystatin-S, S100A11, SMR3B, and immunoglobulins). Changes were already present at the pre-diabetic stage and in T2DM without DR, even in the absence of clinical ocular complications.	National Institutes of Health, USA; Chinese funding bodies	None declared.	Tear proteomic panel (17 proteins, including cystatin-S, S100A11, SMR3B, immunoglobulins)
Jiao et al. [5]	2025	Review	Affiliated Hospital of Shandong Second Medical University; Zhengzhou University People’s Hospital, China	Not applicable	Redefines tear film as an active immune interface. Highlights the role of proteomics, metabolomics, and lipidomics in understanding tear composition. Describes biomarkers linked to immune regulation, mucin deficiency, and systemic disease (including diabetes). Emphasizes challenges such as small sample volume and lack of standardized tear collection methods.	Not reported	None declared	Lactoferrin, Lysozyme, EGF, sIgA, Lipocalin-1, PRG4, mucins (MUC1/4/16/20, MUC5AC), defensins, cytokines (IL-1β, IL-6, IL-17A, IFN-γ, TNF-α), metabolic and lipid markers (OAHFAs, LPCs, triglycerides)

## Data Availability

The data used in the review are available upon request.

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
