# Peer review of "Tear Fluid Biomarkers in Diabetic Ocular Surface Disease: A Systematic Review"

_jcm, 2025, doi:10.3390/jcm14196958_

Round 1

Reviewer 1 Report (Previous Reviewer 1)

Comments and Suggestions for Authors

All my cooments have been addressed

Author Response

Dear Reviewer,
I would like to sincerely thank you for the positive evaluation of my work and for the time you devoted to its careful reading. I greatly appreciate your encouraging feedback.

Reviewer 2 Report (Previous Reviewer 2)

Comments and Suggestions for Authors

  1. Lines 16–27 (Abstract – Results)
    The abstract is descriptive but lacks quantitative detail. Please include the total number of participants across the included studies, the distribution of DM1 vs. DM2, and the main biomarker categories. This would make the abstract more informative for clinicians.
  2. Lines 64–68 (Methods – Registration)
    The lack of PROSPERO registration is acknowledged but underemphasized. This is a significant methodological limitation. Please justify this decision more clearly and explain how selective reporting bias was mitigated.
  3. Lines 93–101 (PRISMA flow diagram)
    The PRISMA flow diagram is mentioned but not shown. Please provide the complete PRISMA 2020 diagram with numbers for identified, screened, excluded, and included records.
  4. Lines 103–106 (Results – Risk of bias)
    Risk of bias is summarized in general terms. Please include a supplementary table showing the JBI checklist assessments for each study, allowing readers to evaluate methodological quality.
  5. Lines 108–147 (Inflammatory biomarkers)
    This section reads more like a narrative review. Please summarize systematically by reporting how many studies found increased versus unchanged levels of IL-6, TNF-α, IL-8, MCP-1, and other cytokines, and highlight consistencies or discrepancies.
  6. Lines 165–215 (miRNAs, circRNAs, VEGF)
    Several studies included here are small pilot investigations. Please emphasize their exploratory nature and group them as “emerging biomarkers” with preliminary evidence rather than established markers.
  7. Lines 231–299 (Proteins and lipids)
    Tables are detailed but not sufficiently integrated into the text. Please reference them explicitly, for example, by stating which biomarkers (e.g., IGFBP-3) showed the most consistent associations and which were less reliable.
  8. Lines 309–335 (Neurodegeneration biomarkers – Substance P, NPY)
    The evidence is presented separately without integration. Please clarify whether Substance P and NPY provide complementary or conflicting information across DM1 and DM2, and indicate which seems most promising for neuropathy detection.
  9. Lines 439–471 (Discussion)
    The discussion section is currently a repetition of results. It should instead compare the findings with previous reviews (e.g., Markoulli 2017, Hagan 2016, Winiarczyk 2022), highlight the strengths of this review (systematic synthesis of multiple biomarker classes, PRISMA adherence), acknowledge its limitations (no PROSPERO, heterogeneity, no meta-analysis, small samples), and propose future research directions (multi-marker panels, validation in larger cohorts, standardized tear collection methods, biosensor development).
  10. Lines 474–483 (Conclusions)
    The conclusions are too general. Please specify which biomarkers show the strongest evidence (IGFBP-3, progranulin, glycan signatures), which are less specific (IL-6, TNF-α, VEGF), and the most realistic clinical applications in the near term (screening, monitoring, risk stratification).

Author Response

Dear Reviewer,
I am sending you all comments with responses in the file.
Yours sincerely,
Natalia Gospodarczyk

This manuscript is a resubmission of an earlier submission. The following is a list of the peer review reports and author responses from that submission.

Round 1

Reviewer 1 Report

Comments and Suggestions for Authors

The manuscript addresses an important and clinically relevant topic. The potential of tear film analysis as a non-invasive diagnostic tool for diabetes-related ocular surface disease is of high scientific and clinical interest.

Below I provide detailed feedback.

The absence of registration in PROSPERO and that no protocol was prepared is a significant methodological limitation. The authors mentioned risk of bias assessment, but the results are not presented. Please consider including a summary of the quality assessment with the discussion of how the risk of bias might affect the overall conclusions.

The title and abstract are focusing on the ocular surface. However, several sections of the results (e.g., sections on miRNA, circRNA, VEGF) address biomarkers in the context of diabetic retinopathy, which is a posterior segment disease. The discussion should link these biomarkers to their presence in tears and show their relevance to ocular surface pathophysiology. Probably the authors could explain the role of these biomarkers as of tear-based proxies of microvascular complications. Alternatively, the title and scope should be broadened to reflect this content.

The Results section is currently quite descriptive, presenting findings from individual studies with limited amount of synthesis. The Discussion section is brief and does not adequately interpret the findings or address the clinical implications. It would be beneficial for the manuscript to group biomarker based on pathological processes such as inflammation, or oxidative stress.

Line-by-line Comments

Line 45 – Please provide a reference.

Line 47 – Please provide a reference.

Line 58 – Please provide a reference.

Line 63 – Please provide a reference.

Line 67 – Please provide references.

Line 76 – Please provide a reference.

Lines 79–80 – For consistency, please use quotation marks when describing terms used in the search strategy.

Line 77 – Only PubMed and Scopus were searched. Please clarify and justify this choice, or expand the search.

Line 94 – Please provide the criteria and tool used for risk of bias assessment.

Lines 110–112 – This section should be deleted

Lines 118–128 – his paragraph discusses the application of HTS in intraocular fluid, which is outside the stated aim of the manuscript. Please reframe the introduction to this section to focus explicitly on the use of HTS in tear fluid analysis for ocular surface disease.

Line 122 – Missing references.

Lines 132–134 – Consider moving this paragraph to the Discussion section.

Line 157 – Missing reference.

Line 160 – Missing reference.

Chapter 3.2 (circRNA) – Similar to the miRNA section, it is unclear whether the findings are from studies on tear fluid or intraocular/vitreous samples. This must be clarified. The relevance to tear fluid and the ocular surface must be explicitly stated.

Lines 205–220 – The authors report detailed quantitative data (means, SDs, p-values, R-values) from the study by Stuard et al. For the sake of balance and consistency, similar detail should be provided for other key biomarkers discussed throughout the manuscript, or this level of detail should be harmonized.

Table 1 – Check consistency in decimal separator (commas vs. points).

Chapter 3.8 – Again, the authors present raw data from an original study. Please summarize findings instead of reproducing exact values.

Line 309 – Missing reference.

Table 2 – The table, in its current form, is not very informative. To be useful, it must include indicators of statistical significance (e.g., p-values) comparing the CON, DM, and DR groups. Without this, the reader cannot assess the importance of the changes reported.

Lines 422 and 425 – Missing references.

Table 3 – Would be more informative if enzyme values in each group and corresponding p-values were shown.

Table 4 – Table formatting is problematic; margins must be adjusted to improve readability.

Discussion – Too brief and lacking interpretation of findings. Please expand to critically assess diagnostic specificity, feasibility, and clinical translation.

Reviewer 2 Report

Comments and Suggestions for Authors

The article addresses an interesting and relevant topic, but it needs major improvements in terms of structure, clarity, and scientific rigor.

1. Introduction

• Lines 38–44: Summarize and avoid repetition. The introduction is too generic and lengthy. For example: Diabetes mellitus (DM) is a chronic metabolic disorder characterized by hyperglycemia resulting from insulin deficiency or resistance. It affects over 415 million adults globally, a figure projected to reach 640 million by 2040 [1].

• Line 44: When mentioning “macro- and microvascular complications”, directly refer to ocular manifestations to focus the article from the start.

• Lines 45–53: The explanation of the tear film and its composition is excessive in this context. Condense into a single sentence and link directly to the importance of tear alterations in DM. The tear film, comprising water, lipids, proteins, and mucins, plays a critical role in ocular surface protection. In diabetes, compositional changes in the tear film may reflect both local and systemic disease.

• Lines 54–65: The biomarker definition is redundant (two definitions are given). Choose one (NIH or FDA), and merge it with the statement about the utility of biomarkers in tears. A biomarker is a measurable indicator of biological or pathological processes. Tears are an attractive medium for biomarker discovery, as they can be collected non-invasively and reflect both ocular and systemic health.

• Lines 71–74: The study objective should appear earlier, ideally at the end of the first paragraph of the introduction. This systematic review evaluates the potential of tear fluid biomarkers in diabetic ocular surface disease.

2. Materials and Methods

• Line 75: Indicate whether the review followed a registered protocol or justify why it was not registered. Although the review adhered to PRISMA 2020 guidelines, it was not registered in PROSPERO.

• Lines 76–85: Better specify the exact search terms and the specific dates (month/year) when the search was conducted. Explain inclusion/exclusion criteria before the search methods.

• Lines 86–105: Standardize the description of study selection and data extraction processes. For example: Two independent reviewers screened titles and abstracts; disagreements were resolved by consensus or by a third reviewer.

• Line 105:There is a lack of detail on risk of bias assessment: what tool was used? Was the quality of evidence assessed?

3. Results

• Lines 110–112:Delete generic phrases such as “This section may be divided by subheadings.”

• Subsection structure (miRNA, circRNA, Glucose, Proteins, etc.):Begin each subsection with a brief introductory statement on the relevance of the biomarker. Indicate limitations or inconsistencies in the results at the end of each subsection (where applicable). Add a comparative synthesis in a final sentence for each subgroup.

• Lines 113–149 (miRNA, circRNA):Avoid long lists of miRNAs; summarize with: Several miRNAs, including…, were found up- or downregulated (see Supplementary Table X if applicable).

• Lines 170–177 (Tear glucose):Emphasize the controversy and lack of correlation, and suggest caution in interpretation.Tear glucose concentrations in diabetics are inconsistently correlated with blood glucose, likely due to methodological variability and contamination. Thus, their clinical utility remains uncertain.

• Lines 178–219 (Proteins):Summarize proteomic findings in a table (see below), highlighting only the main findings in the text. Add a comment about DED as a frequent confounder.

• Lines 221–227 (Meibum Lipids): State which studies and sample size; are these changes clinically relevant?

• Lines 228–246 (Metallic Elements): Summarize clinical relevance in a final sentence and avoid excessive technical detail.

• Lines 251–279 (Substance P and NPY): Summarize, note differences between type 1 and type 2 diabetes (do not repeat phrases for both), and indicate sample size limitations.

• Lines 305–337 (GAGs, Glycans, Glycated albumin):For glycans/glycation, summarize their relevance versus HbA1c and specify whether they are useful for DM1, DM2, or both.

• Lines 359–399 (Cytokines, VEGF):Highlight the low specificity of inflammatory markers (e.g., TNF-α, IL-6) due to comorbidities such as DED; suggest the need for multi-marker panels.

• Lines 409–416 (Nerve density and length): State which methods (e.g., confocal microscopy) were used and whether they are applicable in routine practice.

• Lines 417–438 (NETs):Summarize diagnostic/therapeutic potential in one sentence; clarify preclinical vs clinical evidence.

Lines 439–453 (Oxidative Stress): Avoid unnecessary technical terms and tables in the main text; move these to the supplementary section.

• Lines 454–470 (Progranulin): Summarize evidence and highlight sample size limitations and need for validation.

4. Discussion

Include a narrative discussion synthesizing the main findings of the study, comparing them with previous literature, and reflecting on the clinical and scientific relevance of the identified biomarkers.

Clearly state the study’s limitations, both methodological (heterogeneity, bias, lack of longitudinal studies, etc.) and clinical (lack of external validation, low specificity of some biomarkers, etc.).

Propose future research directions and discuss the potential applicability of biomarkers in clinical practice.

Present the summary table as supplementary material or in a “Summary of included studies” section, but not as a substitute for the discussion.

5. Conclusions

1. The tear film is a promising medium, but biomarkers require further validation.

2. Methodological and outcome heterogeneity limits immediate clinical implementation.

3. Progress towards multi-marker panels and standardized sampling procedures is necessary.

Avoid repeating text from the introduction/discussion.